# The Rheology and Physicochemical Characteristics of Hyaluronic Acid Fillers: Their Clinical Implications

**DOI:** 10.3390/ijms231810518

**Published:** 2022-09-10

**Authors:** Salvatore Piero Fundarò, Giovanni Salti, Dennis Malvin Hernandez Malgapo, Silvia Innocenti

**Affiliations:** 1Multimed Poliambulatorio e Day Surgery, Via dei Fornaciai 29/d, 40129 Bologna, Italy; 2Medlight Institute, Via Monteverdi 2, 50144 Florence, Italy; giovannisalti@gmail.com; 3Skinsoul Advanced Dermatology and Medical Aesthetics, Spark Place P. Tuazon Blvd, Cubao, Quezon City 1109, Metro Manila, Philippines; dmhmalgapo@gmail.com; 4RELIFE S.r.l., 3, Via dei Sette Santi, 50131 Florence, Italy; sinnocenti@relifecompany.it

**Keywords:** hyaluronic acid fillers, rheology, physicochemical properties

## Abstract

Hyaluronic acid (HA) fillers have become the most popular material for facial volume augmentation and wrinkle correction. Several filler brands are currently on the market all around the world and their features are extremely variable; for this reason, most users are unaware of their differences. The study of filler rheology has become a wellspring of knowledge, differentiating HA fillers, although these properties are not described thoroughly by the manufacturers. The authors of this review describe the more useful rheological properties that can help clinicians understand filler characteristics and the likely correlation of these features with clinical outcomes.

## 1. Introduction

As a result of our deeper understanding of facial aging and the concomitant increase in soft tissue fillers available on the market, the use of these medical devices has evolved. Minimally invasive procedures for the correction of age-related defects on the face have become the norm, not only for superficial soft tissue, such as the skin and subcutaneous tissue, but also for deep anatomical layers of the face. Widespread use of soft tissue fillers has been attributed to the introduction of hyaluronic acid (HA) fillers, now comprising about 80% of all fillers used for rejuvenation and volume correction [1]. HA fillers have been reported to have low complication rates, good durability [2], are relatively inexpensive, and can be corrected through lysis by hyaluronidase injection [3]. HA fillers are a hydrogel made of crosslinked HA, suspended in physiological or phosphate-buffered solution. The most common crosslinker is 1,4-butandioldiglycidyl ether (BDDE) [4], but other crosslinkers have been introduced [5], namely 1, 2, 7, 8-diepoxyoctane (DEO), divinyl sulfone (DVS), hexamethylenediamine (HMDA), and polyethylene glycol diglycidyl ether (PEGDE) [6].

The HA used for soft tissue fillers is typically obtained from either avian sources, as in from rooster combs, or from bacteria-sourced HA, through the synthetic fermentation of *Staphylococcus equine*. Most modern HA fillers are derived from bacterial HA because of its reduced allergenic and immunogenic potential (i.e., HA from animal sources may retain impurities that could cause adverse reactions) [7]. Different manufacturing procedures provide means to alter HA concentration, crosslinking degree, reabsorption, biophysical, and rheological characteristics to ultimately fit different clinical indications. For this reason, not all fillers are the same and clinicians, in their daily practice, select different products for distinct indications, depending on their personal clinical experience. The selection is based mainly on the direct use of the products, on personal experience, and on the indication provided by the manufacturers through commercial and educational activities.

A literature search of the Medline^®^, Embase^®^, and Google Scholar databases was performed using the search terms “dermal filler”, “hyaluronic acid”, “rheology”, “characteristics”, “physicochemical properties”, and “crosslinking”. The material reviewed included recent meta-analyses, reviews, rheological, and biophysical studies. The literature search included journal articles published from February 2009 to June 2022. The literature review was implemented based on references found in the articles selected in the search. Subsequently, the authors reviewed the literature, identifying the more frequent biophysical and rheological characteristics described in scientific articles and analyzed their method of measurement, definition, practical meaning, and clinical implication.

## 2. A Review of Rheology and Physicochemical Characteristics of HA Fillers

### 2.1. Rheological and Biophysical HA Filler Characteristics

Clinicians are encouraged to understand the rheological and physicochemical properties of fillers (Table 1) to facilitate proper HA filler selection [8]. In this review, we identify nine characteristics with significant clinical implications in filler science.

Understanding the fundamentals of each rheological and biophysical feature and of their clinical implications facilitate the choice of the correct HA filler for each specific use and appropriate injection plane. It is generally accepted that a filler used in the deep plane for facial volume restoration has different characteristics compared with a filler used for fine lines of the skin. Fillers for deep injection are generally defined as “harder” and fillers for fine lines as “softer”. Soft fillers are thought to have lower viscosity and elasticity and have the tendency to spread into soft tissue (i.e., ideal for fine lines and wrinkles). Hard fillers, on the other hand, have higher viscosity and elasticity and provide lift and support, with negligible product migration (i.e., ideal for volume restoration) [14]. Though descriptive, these two terms are not able to detail the behavior of the filler after injection and its interaction with deforming forces that act on it. After injection, fillers are subjected to compression, shearing, stretching, torsion for muscle movements, soft tissue weight, pressure on external surfaces (e.g., on a pillow), and gravitational force. All these forces modify the shape, distribution, duration, and grade of correction in the defect from the injected filler.

Moreover, each filler is described by manufacturers for the same indication without considering that the rheological and physicochemical characteristics are significantly different among filler brands, i.e., fillers may share the same indications while having different rheological, physical, and chemical features. An example of such a type of dissimilarity exists between two big families of HA fillers: the “monophasic” and the “biphasic” (also known as “cohesive” and “granular”) fillers [12,14,15]. Evidently, the monophasic filler is a homogeneous blend of crosslinked HA chains with high or low molecular weight and the biphasic type contains reticulated HA particles dispersed in a vehicle (non-crosslinked or very low crosslinked HA) that act as a fluid matrix that allows the gel to be injected (Figure 1) [16]. These two HA filler types have different modalities of production that lead to dissimilar rheological and physical characteristics while sharing the same indication. Generally, monophasic fillers have lower elasticity and higher viscosity than biphasic HA fillers [17]. Inappropriate use of these filler types may lower the quality of the final aesthetic results.

### 2.2. The Manufacturing Technology of Fillers

The naturally occurring linear form of hyaluronic acid molecules is rapidly degraded by hyaluronidase and, because of its short half-life, may be insufficient to provide satisfactory soft tissue filling. It is, thus, necessary to modify the physical properties to increase HA molecule resistance to resorption. To reach this goal, the polymerization of HA is increased by a crosslinking process that adds a molecule, bridging the polymer chains to one another [18]. This modification process has been referred to as *crosslinking*, *reticulation*, or *stabilization*. Crosslinked HA, being less susceptible to chemical and enzymatic hydrolysis, shows a prolonged in vivo persistence [19] because the HA solution becomes less viscous as it transforms in a visco-elastic gel. This creates a steric barrier that reduces the penetration and the mobility of hyaluronidase inside the gel, increasing the longevity of the HA filler in the soft tissue. The grade of crosslinking contributes to the “hardness” of HA gel, increasing the crosslinking grade. This process makes it possible to augment the rigidity of the gel to the point of being a solid material. For this reason, the crosslinking process highly influences the physical and rheological characteristics in HA fillers.

The most frequently used crosslinker is BDDE, which provides irreversible covalent bonds between HA chains. The epoxide groups present at the two ends of the BDDE molecule preferentially react with the nucleophilic groups of HA, forming an ether bond [20]. BDDE has lower toxicity than other crosslinking molecules, creates a stable three-dimensional network [21], is easily biodegradable, and has been well explored in various studies. Although a slight mutagenic action has been detected in Drosophila [22], no definitive carcinogenic effect has been observed in more complex organisms (i.e., mice) [20]. Because of this potential mutagenic action, the quantity of unbound BDDE molecules in HA filler must be maintained under determined levels. The Food and Drug Administration (FDA) have recommend a residual level of unreacted BDDE of <2 parts per million (ppm) to be safe. This would equate to <0.002 mg of BDDE in 1 mL of HA gel [23].

During the crosslinking process, the epoxide groups of BDDE react with nucleophiles forming derivatives of 1,4-butanediol di-(propan-2,3-diolyl)ether (BDPE). Some portion of the added BDDE reacts only with water/hydroxide, forming free BDPE. Another portion of the BDDE reacts with water/hydroxide at one end and with HA at the other end, forming mono-linked BDPE. A third portion of the BDDE reacts with HA at both ends, yielding disubstituted BDPE, resulting in the cross-linkages found in the HA hydrogels [24] (Figure 2). The free BDPE and the mono/linked BDPE are not useful for the stabilization of HA filler and increase the quantity of BDDE molecules, without any functional effect. For these reasons, the crosslinking technique used by the manufacturers must assure high effectiveness in the crosslinking process and reduce the percentage of free and mono-linked BDPE.

Different parameters may be used to describe the grade and effectiveness of crosslinking of an HA filler [11,24]. Some of these measures are useful to describe specific characteristics. Apart from crosslinking, the mechanical and physical properties of the hydrogels are also dependent on the degree of modification [26]. The relationship between degree of modification and of crosslinking to mechanical properties as well as to biocompatibility is of great importance and, as such, there are several methods that are used to describe this in literature [23,27]. Degree of modification is the stoichiometric ratio between the sum of mono- and double-linked BDPE residues and HA disaccharide units and provides the total amount of linked BDPE in comparison to the total amount of HA. All BDPE molecules, both mono linked and double linked to HA, are included in the calculation of degree of modification. A gel with a low degree of modification will resemble the intact polymer, while a gel with a high degree of modification is a highly modified polymer. The HA molecule, being identical across species, is not recognized as a foreign material after its injection into the human body. It is important not to modify the HA by the crosslinking process to such a great degree that it will not be recognized as HA and lead to a foreign body reaction. The degree of modification indicates the tendency of the HA filler to stimulate a reaction by the immunological system and to be perceived as a foreign material [11]. Edsman and colleagues showed that the degree of modification of 13 investigated products from seven different manufacturers varies from 1% to 8%. Three biphasic HA fillers have a lower degree of modification (1%); the other 10 monophasic fillers have a higher degree of modification (from 4% to 8%). Other authors have confirmed this difference between the degree of modification of biphasic fillers and monophasic fillers [13].

Another ratio, the crosslinking ratio, relates the crosslinked BDPE to the total number of BDPE molecules bound to HA. An HA filler modified with predominantly suspended BDDE molecules has a lower crosslinking ratio and is weaker than an HA filler with predominant double-linked BDDE that has a higher crosslinking ratio and is more strongly attributed to the efficient covalently crosslinked network. The degree of crosslinking, which is the stoichiometric ratio between BDPE residues that are double linked and HA disaccharide units, provides the total amount of double-linked BDPE, in comparison to the total amount of HA. Degree of modification and degree of crosslinking describe different properties and, hence, have different implications. For example, higher values of degree of crosslinking or of crosslinking ratio reveal that a gel is stronger because it is more crosslinked and would swell less than a weaker gel with a lower degree of crosslinking or crosslinking ratio. The greater the degree of crosslinking, the harder the gel becomes. The higher the degree of crosslinking, the longer the persistence of the filler after injection in soft tissue. The higher the degree of crosslinking, the lower the filler’s hydrophilicity [23]. The heterogeneity in the available methods measuring these parameters does not allow for comparison of data obtained from the different manufacturers. Currently, no agreement about the method used to calculate these parameters exists.

Each manufacturer describes a proprietary crosslinking technology, trademarked names, and supposedly peculiar physicochemical characteristics of their HA filler. Table 2 summarizes the name of crosslinking techniques used by the main filler brands.

When the crosslinking phase is completed, the result is a compact gel mass that requires further processing. The gel mass must then be “sized” into smaller crosslinked domains to reach a suitable viscosity for injectability. Tezel and Fredrickson [23] describe two different methods to reach an appropriate “sizing” of gel. The first method consists of passing the gel mass through a series of sieves or screens. Through this method, gel particles of a well-defined average size are created. Different products have distinct average gel particle sizes according to the proprietary sieving method applied during the manufacturing process. Gel particle size also determines the “hardness” of the gel: the bigger the size, the harder the gel.

There is a maximum particle size, beyond which gel particles would not extrude through a needle. To facilitate the injectability, particles of the desired size are dispersed within a soluble HA phase that acts as a lubricant. Fillers produced in this way are commonly described as “biphasic” [32], “granular”, or “particulate” [7]. An alternative method to size the gel mass is homogenization. The result is a gel with a smooth consistency and with a more regular surface compared with the more granular consistency of gel particulate formulations previously described. The particles created by homogenization have a wide array of sizes [12]. Fillers produced in this way are commonly described as “monophasic” [32], “smooth” [33], “non-particulate” [34], “homogenous” [35], “continuous” [36], or “cohesive”. In recent years, the terminology commonly used to describe the filler produced by these two different methods has undergone scrutiny and different opinions have been expressed by several authors. Ohrlund et al. maintained that both types of filler are particulate, indeed, using the dispersion and staining technique; hence, the particulate nature of any crosslinked HA filler on the market can be demonstrated [12]. After dispersion in water and staining with toluidine blue, the gel particles are easily discernible in products from both product families [37]. Nonetheless, the authors acknowledge that the major difference between the product families is the particle size distribution resulting from the different types of particle-sizing processes used. Certain authors propose that all fillers must be considered monophasic because HA fillers have the same composition throughout. Sundaram and Cassuto [38] state that it may be more accurate to think of homogenized cohesive HA fillers as biphasic and to think of sieved HA filler as behaving in some respects like a triphasic gel. Since there is no univocal acceptance of nomenclature, we propose to name these two different types of HA filler as “homogeneous particle fillers” and “inhomogeneous particle fillers”. These definitions highlight the main difference unanimously recognized by several authors.

Definitions are necessary according to another school of thought [11,23,33,39]. Specific production methods determine different physical and rheological characteristics: the “homogeneous particle fillers” are harder, allow for less flow, and are less cohesive than “inhomogeneous particle fillers”. The different types of HA fillers have different behaviors immediately after their injection in the dermis and show different histological distribution patterns that are consistent between patients that are somehow predictable [32,40].

### 2.3. Rheology of Fillers

Proprietary manufacturing processes are used to alter HA molecular structure as well as their physicochemical and mechanical behaviors. These varied behaviors lend individual agents their unique rheological characteristics and are perceived to affect their overall product performance. Rheology (/ri:′ɒlədʒi/; from Greek ῥέω rhéō, ‘flow’ and λoγία, -logia, ‘study of’) is a branch of physics that deals with the deformation and flow of liquid, gaseous, and soft solid (like gel) matter [41]. The term, inspired by the aphorism of Simplicius (often attributed to Heraclitus), panta rhei (πάντα ῥεῖ, ‘everything flows [42], was coined by Eugene C. Bingham [43]. It studies the behavior of materials when subject to deforming forces and applies to substances that have a complex microstructure, such as mud, suspensions, topical medication, as well as paints, inks, industrial and mechanical oils, bodily fluids (e.g., blood), and other materials that belong to the class of soft matter.

The knowledge and the understanding of the rheological properties in HA fillers can help in the selection of products by clinicians and the identification of the best suited for each indication, facial region, and anatomical layer. Soft tissue tension, muscle movements, gravity, and pressure on external surfaces (a pillow during sleeping) apply on HA fillers several types of forces that determine the shear deformation, vertical compression, and stretching. Each of these forces varies based on the depth of injection, on the area of the face, and on the types of mimetic movements that each zone presents. Therefore, it is necessary to use fillers with different rheological characteristics that react adequately to the applied forces to achieve optimal defect correction [44]. This cannot be done without knowing the significance and the clinical implications of the main rheological features of HA fillers [14,28,45] (Table 3).

Rheologic features are typically measured using rheometers. The main category of rheometers concerned with rotational or shear rheometers relies on rotational motion to achieve a shearing effect. The first method was introduced in 1888 by Maurice Couette and the second method was introduced in 1912 by George Searle. Using the two methods of measurement, rotational rheometers can be classified into four different categories. A simple dynamic shear rheometer, or DSR, can be used to measure the characteristics of a wide range of materials under varying conditions of temperature, stress, and strain. Rotational cylinder rheometers have two cylinders, one inside the other, where the inside rotates at a known speed for calculating shear stresses on a sample. A pipe or capillary rheometer runs liquid through a tube with set dimensions and flow rates to calculate the shear rate on the tube. Finally, the last type of rheometer, parallel-plate or cone and plate rheometers, use rotating plates or a shallow cone and plate with liquid between them to measure the shearing forces on the sample [46]. The results obtained are always dependent on the used device and cannot be compared with each other. Lorenc et al. tested the rheological properties of eight HA fillers and they concluded that a large part of the differences are due to differences in rheometric measurement settings and that, since analytical results are always influenced by instrument settings, consensus on settings is essential to make the comparison of results from different investigators more useful [47]. The heterogeneity in the rheological data found in the literature imposes a challenge for clinicians to extrapolate useful indications to determine their filler choice. The standardization of used measurement methods, including the use of various rheometers (e.g., capillary rheometer) and selection of main and more relevant rheological parameters usable for filler selection may facilitate the comprehension and the practical use of a filler’s rheological characteristics by the clinicians.

Before describing the main rheological characteristics in HA fillers, it is necessary to introduce some basic concepts of physics. Matter can exist in four fundamental states: solid, liquid, gas, and plasma. The last three states are defined as “fluid”; however, gas and plasma are not of interest in the discussion of rheology in HA fillers. We can define solid as a material having fixed volume and shape and liquid as a material having a volume but not a shape. The behavior of solids and liquids differs for a lot of rheological characteristics. First, solids have a shape and liquids do not. The ability to maintain a shape is the rigidity and, as solids are rigid and fluids are non-rigid, they can flow. The property of rigidity in rheology of solids is the measure of elasticity, which can be defined as the ability of a material to resist deformation and it is determined by Hooke’s law, which states that the elastic strength of a material (the elastic modulus) is equal to the ratio of the stress applied to the material over the strain that is induced in the material.
Elastic modulus = Stress/Strain

To better understand the concept of stress, we can use an example: when a solid is exposed to an external compressive force, the molecules of the matter are pushed together and they accumulate a repulsive force. The internal pressure, determined by the repulsive force, is the stress. We can define stress as the internal pressure that material is subject to when external forces are applied. Stress, being a pressure, is measured in Pascals (Pa). The relative deformation due to the external force is the strain and it is a measure of how much the dimension of an object has changed. Forces can be applied in different ways on an object. If the forces compress or stretch an object, aligned one to another, they are defined as “normal stress”. If the forces act not aligned, they cause the object to change form but not its volume and these forces are collectively called “shear stress”. If the normal stress acts in a three-dimensional way and it compresses or stretches the object in all directions equally, this is referred to as “volumetric stress”. Each of these three stresses has a specific elastic modulus: normal stress is associated with the elastic modulus (E), shear stress has shear elastic modulus (G), while volumetric stress has bulk elastic modulus (K).

The elastic modulus (E) is the ratio of normal stress (σ) over the normal strain (ε)
E = σ/ε
and it represents the tensile strength or compressive strength. It is higher if the object is less compressible or stretchable.

The shear elastic modulus (G) is the ratio of shear stress (τ) over the shear strain (γ)
G = τ/γ
and it reflects the elastic strength of a solid material. As G is higher, the material is stronger and less deformable. Finally, the bulk elastic modulus (K) is not relevant in HA filler rheology.

Fluids are shapeless so they are unable to resist deformation and, for this reason, their E and G are essentially zero. Fluids have an intrinsic and specific characteristic: viscosity (η). Viscosity can be defined as the ability of a fluid to resist flow. It is described by Newton’s law of viscosity: it is the ratio of shear stress (τ) over the shear stress rate (γ•), which describes how quickly a fluid is flowing. The unit for viscosity is in pascal-seconds.
η = τ/γ•

Viscosity indicates the pressure necessary to determine the flow of a fluid. To better explain this concept, we can compare water and honey: if we want to extrude water and honey through a syringe, we need to apply less pressure when we extrude water versus honey. This means that water has a lower viscosity than honey and we can state that water is less viscous than honey. Also, water is a Newtonian fluid, while honey is a non-Newtonian fluid. In this last fluid type, viscosity can change when under force to either more liquid or more solid. Ketchup, for example, becomes runnier when shaken and is, thus, a non-Newtonian fluid. HA fillers are classified as non-Newtonian fluid. When pushing the plunger in the syringe, HA feels hard, but when pressure is increased, it suddenly starts to flow easier and becomes less viscous.

Materials may exist in the solid phase and in the fluid phase, but we can create a mixture, or a dispersion, combining the two phases. It is possible to generate three different dispersions: the solution (i.e., water and salt) where the particles are <1 nm, the suspension (i.e., water and sand) where the particles are >1 μm, and colloid, such as milk, in which the dispersed particles, between 1 nm and 1 μm, are too big to be dissolved and too small to precipitate but are homogeneously dispersed. If we create a colloid mixing a liquid and a solid, we create a gel and, if the liquid is water, the gel is defined as hydrogel. If we mix hyaluronic acid powder and water, we obtain a hydrogel with solid and fluid components and, therefore, this presents both the main characteristics of these two states: the elasticity in the solids and the viscosity in the fluids.

HA fillers exhibit both elastic and viscous behavior and viscoelasticity is their main characteristic. To better understand the concept of viscoelasticity, we may describe a rubber band as a purely elastic matter that deforms up to a certain point under shear stress and recovers its original shape when the force causing deformation is removed. The rubber band is an example of an elastic material, which stores energy that it will use to restore its strain. Honey is a purely viscous matter that is not able to recover its shape when the force causing deformation is removed. Honey is an example of viscous material with the ability to lose the energy that should enable it to recover strain. A purely elastic matter cannot be injected into the soft tissue because it is too hard and a purely viscous matter is not able to restore the soft tissue volume because it is too liquid. The balance between these two characteristics in the HA filler determines its rheological features and makes it ideal for specific indications.

The viscoelastic characteristics in HA fillers are described using five main rheological parameters: the elastic/storage modulus (G′), the viscous/loss modulus (G″), the complex modulus (G*), tangent delta (tan δ = G″/G′), and complex viscosity (η*). In general, it is important to note that the viscoelastic properties of materials are reported through particular amplitudes and time scales and these parameters should, thus, be taken into consideration when evaluating soft tissue filler properties [48].

#### 2.3.1. The Elastic/Storage Modulus (G′)

The elastic modulus is a measure of the energy stored in a material, in which shear deformation has been imposed. In other words, elastic modulus can be thought of as that proportion of the total rigidity (the complex modulus) of a material that is attributable to elastic deformation. It represents the energy fraction of G* that the gel can store during shear deformation and that can be used to recover its original shape when deformation is removed. Basically, HA fillers with a higher G′ are firmer, with a more elastic response to shear deformation, whereas lower G′ products are softer and less elastic [11,47]. Generally, G′ is used to predict and describe the lift capacity in the fillers, even if, recently, this seems true among HA fillers with the same composition/crosslinking technology. For instance, fillers manufactured via XTR^TM^ Technology have relatively higher G′ than other HA fillers for corresponding indications and are, hence, believed to have lasting effects and more pronounced lifting capacity [31]. Additionally, it is likely that the lift capacity is not correlated only with G′, but is also linked with other parameters that influence the fillers’ lifting performances, especially when the fillers are injected and inserted into live soft tissue [40]. In any case, G′ is an expression of the sum of numerous features that affect filler strength as the hyaluronic acid concentration and the degree of crosslinking; therefore, G′ has become a relevant parameter used to describe the firmness and volumizing capability of products [49].

Because G′ describes elasticity when shear forces are applied but not when normal forces act on the filler, to measure this type of elasticity, the elastic modulus E′ is described. It is detected with the rheometer in compression oscillation mode, applied at different frequencies. It represents the ability of the product to resist dynamic compression and indicates the gel’s capacity to recover its shape after a vertical compression. This rheological feature is proposed as a useful rheological parameter by a handful of authors [50], but it has a significant role in the filler behavior description, completing and integrating the information provided by G′.

#### 2.3.2. Viscous/Loss Modulus (G″)

The viscous modulus is a measure of the energy dissipated in a material, in which deformation has been imposed. It can be thought of as that proportion of the total rigidity (the complex modulus) of a material that is attributable to viscous flow, rather than elastic deformation. G″ may be described as the energy fraction of G* lost on shear deformation through internal friction or, in other words, it is the measure of a gel’s ability to dissipate energy when shear force is applied to it [38]. Like complex viscosity (η*), G″ measures the viscosity when shear force stays within the linear viscoelastic region (LVE region), which indicates the range of the forces in which the test can be carried out without destroying the structure of the sample. G″ is related to G′ and they, within the LVE region, are used to indicate the viscoelastic character of the gel. If G′ > G″, then the gel shows a gel-like or solid structure and can be termed a viscoelastic solid material. However, if G″ > G′, the sample displays a fluid structure and can be termed a viscoelastic liquid. All HA fillers have a G′ > G″, which means that they have a gel-like structure and are viscoelastic solid materials. Even if G″ is related to viscosity, researchers prefer measuring this characteristic using η* because it gives a better representation of how a filler might be affected by shear forces during and after injection [38]. G″ may not be an accurate indicator of viscosity because HA fillers are not purely viscous [51].

#### 2.3.3. Complex Modulus (G*)

This measures the total energy needed to deform a material using shear stress. It indicates the overall resistance to deformation of a material, regardless of whether that deformation is recoverable (elastic) or non-recoverable (viscous). Complex modulus is a useful property to quantify the gel hardness, as it is a direct measure of the rigidity in a material’s soft solid structure (as a gel) when exposed to stresses below the yield stress. For this reason, it is a good indicator of visible attributes, such as the stiffness or hardness of the HA filler: the higher the magnitude of the complex modulus, the stiffer the material [38,52]. According to Pierre et al. [51], this variable represents how difficult it is to alter the shape of an individual crosslinked unit of filler. G* reflects the “hardness” of multiple units of crosslinked HA, not the hardness of the whole gel deposit. A specific formula allows one to calculate the G*
G* = √((G′)^2 + (G″)^2)
based on the value of G′ and G″ obtained from testing with rheometer. In most HA fillers, the G* has a value approximately equal to G′ because, at the shear forces that act on them in facial soft tissue, they have a low G″. It is for this reason that this value is often used to describe the filler hardness, whereas G′ is commonly utilized. Nonetheless, it may still be considered as a useful parameter because the hardness of the fillers has several implications concerning their clinical use. Hardness may influence the palpability of the filler and because of this, a filler with high G* is more suitable for implants in the deep anatomical planes, such as the supraperiosteal layer and deep-fat compartments. A stiff filler cannot be injected into the dermis because it alters the pliability of the dermis, causing unnatural, visible, and palpable dermal modifications.

#### 2.3.4. Tan Delta (tan δ)

G* is derived from both elastic (G′) and viscous (G″) moduli. The ratio of these components (G″/G′) gives the tangential delta (tan δ). Tan δ, being the result of the ratio between viscosity (expression of fluidity) and elasticity (expression of rigidity), indicates the solid behavior of the material (i.e., more jelly like) or if it exhibits more a liquid-like behavior (i.e., honey like) [51]. Since HA fillers have higher G′ than G″, the ratio is always <1. This means that, generally, HA fillers are mainly elastic and that the lower tan δ is, the more solid or jelly like the filler is. This parameter indicates, in inverse proportion, the elasticity of HA filler and is a good indicator of whether the filler may be injected more superficially (i.e., higher tan δ) or deeper (i.e., lower tan δ).

#### 2.3.5. Cohesivity

Cohesivity is described as the force between particles in the same substance that acts to unite them [53]. In the case of the fillers, cohesivity is an expression of the internal adhesion forces holding together individual crosslinked HA units that compose the HA gel [51]. A material with low cohesivity has particles that easily separate, while it is more difficult in a material with high cohesivity. The scientific opinions concerning the clinical relevance of cohesivity are conflicting; some authors believe that it is related to the filler’s resistance to vertical compression/stretching and with filler projection capacity. A filler subjected to a vertical force is more prone to be divided into smaller particles if it has low cohesivity and has low internal holding forces [15,51]. Conversely, others state that cohesivity does not give any advantage in soft tissue lifting [52] while others posit that cohesivity contributes more to tissue expansion than to projection with a predominantly horizontal vector [54].

There are no ready-made instruments designed to measure the cohesivity and there is no standardized methodology that is validated by the scientific community as an accepted method of measuring cohesivity. The most popular methods are the Gavard-Sundaram Cohesivity Scale [54], the linear compression test, the dye diffusion test [15], and the average drop-weight method [53]. These tests are not known to provide consistent data and in a comparison between different methods, the average drop weight seems to closely resemble the HA filler cohesivity [53]. Several authors [31,49,53] described an inverse correlation between G′ and cohesivity: fillers with higher G′ have lower cohesivity and fillers with lower G′ have higher cohesivity. Moreover, this inverse correlation may have been the underlying reason behind the inaccurate observation made by some authors where more cohesive fillers integrate better into the tissue [32,55]. Apparently, the cohesive behavior of HA fillers with low G′, owing to their softness, allows them to deform more easily and infiltrate the soft tissue better. Therefore, their capability to spread within the tissue in a more homogeneous way is related more with the low G′ than with the high cohesivity [53].

#### 2.3.6. Viscosity

Viscosity indicates the gel’s ability to resist shearing forces, which are applied on a filler during its injection and when administered inside soft tissue. It measures the filler’s resistance to flow when shear stress is applied and the force needed to inject the filler. Since HA fillers are considered as non-Newtonian fluids, their viscosity decreases when the applied shear force reaches a level beyond which the viscosity reduces. For this reason, when we start to inject an HA filler, we perceive a high resistance to flow, until increasing the pressure on the plunger. At this point, we reach the “shear thinning point” and the filler can be injected more easily. If the shear force reaches a determined level beyond the LVER, the physicochemical structure of the gel is disrupted and η* modifies in a dramatic and uncontrolled way. This means that we have reached the gel’s yield stress and that it has no more viscoelastic behavior.

HA fillers with low η* have a low “shear thinning point” and low yield stress: they are easier to inject. However, if they are injected through a thin needle or if high or prolonged forces are applied on them, they reach the yield stress and can lose the viscoelastic characteristics. On the other hand, HA fillers with high η* have a high “shear thinning point” and high yield stress: they are certainly harder to inject and their resistance to forces is stronger.

There is no concordance on the role of viscosity after injection in the scientific community. Pierre et al. maintained that, because of the low shear rates applied on fillers after injection, they exhibit predominantly elastic properties [51]. Therefore, viscosity is not relevant to performance after the filler has been implanted. In the histological study by Flynn et al. [32], they reported that fillers with lower viscosity and elasticity have less clumping and more homogenous staining, while Sundaram et al. [14] maintained that fillers with lower viscosity and elasticity spread homogeneously into the tissue and, thus, have a softer feel and lower lifting effect. Viscosity appears to better define HA filler injectability, interpreted as extrusion from the syringe and immediate tissue integration; meanwhile, the filler behavior after injection into the tissue is better described using other rheological parameters, such as G′ and cohesivity.

#### 2.3.7. Normal Forces

An often unrecognized but still important parameter in filler rheology is compression (E′, tanδc, normal force F_N_) [50,56]. Fillers exhibit constant compression forces, such as when fillers exert mobile compression during movement of facial mimetic muscle, after implantation. These measurements are important because of the direct mechanical effect of fillers on skin tissue. A study by Molliard et al. hypothesized that gels with higher compression forces have a better ability to mechanically stimulate the synthesis of de novo collagen by altering the configuration of tissue fibroblasts. These cells are then stimulated to create structural support around the area of injection, such that it may help reduce the signs of aging in the dermal environment [50].

#### 2.3.8. Thixotropy

Thixotropy is defined as the property of viscous or gel-like products to turn into a liquid as time progresses and more rigid as it is deformed (i.e., as in the process of stirring). It is a term used to denote a change in apparent viscosity under shear stress, followed by a gradual recovery when the stress is removed [57]. After an initial disruption in HA molecular chains, the three-dimensional network assumes its initial structure [58]. Although this represents another important yet underrecognized rheologic property that may differentiate HA-based soft tissue filler technologies, more studies are needed to elucidate its clinical correlation and applications in soft tissue filler injection and tissue integration [59].

### 2.4. Physicochemical Properties

#### 2.4.1. Hyaluronic Acid Concentration

All manufacturers declare the HA concentration (mg/mL) of the HA fillers; this refers to the total HA present in the formulations without specifying the amount of the insoluble crosslinked HA and of the non-crosslinked HA soluble fraction of the biopolymer. The soluble fraction is usually added to optimize the viscosity and improve the product extrusion through a needle of proper dimension or may originate from the small fragments of HA generated during the crosslinking process or the sterilization step [26,33,60]. It is easily metabolized and supposedly does not have any influence on the filler’s characteristics involved in performance and effectiveness (i.e., injectability, spreadability, lifting capacity, duration, etc.) [61]. The soluble fraction of the biopolymer may vary significantly among different fillers and the declared HA concentration in commercial fillers may not directly correlate to the final gel behavior [62]. Usually, physicians are not informed about how much soluble HA is included in the filler and, for this reason, the total concentration of commercially available HA fillers can only be a reference value rather than an absolute parameter for assessing filler performance [44].

#### 2.4.2. Molecular Weight and Polydispersity

The molecular weight (MW) of HA lends structural and physicochemical integrity to fillers. As MW increases, there is an apparent reinforcement in its three-dimensional network. In effect, the higher the MW, the higher the viscosity and viscoelasticity in the filler are [58]. The MW of HA used in the production of soft tissue fillers can range from 500 to 6000 kDa. The sodium salt of hyaluronan often comes as a disaccharide, with an MW of approximately 401 Da. Though MW is used to describe HA gels, a typical filler is made up of crosslinked HA molecules; hence, the actual MW of an HA gel is larger than reported. As a result, minute differences in the MW of the component HA have a negligible effect on the final properties of the gel. Therefore, the number of crosslinks and the percentage of modification essentially determine the characterization of HA gels [13]. Of note, larger-sized HA fillers may display elastic moduli and viscous moduli that are adequate for biphasic fillers [63]. However, MW appears to have no impact on inflammatory/immune response to fillers, regardless of HA crosslinking [64]. Lastly, polydispersity refers to the heterogeneity in the sizes of the molecules in a sample. HA fillers used for medical applications are preferably low-polydispersity or monodisperse HAs [58]. Major filler brands have a similar distribution width or polydispersity index [61].

#### 2.4.3. Swelling Factor or Hydration Capacity

In physiological conditions, water forms hydrogen bonds with the N-acetyl and carboxyl groups and this results in HA’s affinity for retaining water. The swelling factor is determined by adding, while stirring, 0.9% NaCl solution to a certain filler amount. After centrifugation and supernatant removal, the gel volume is measured. The swelling factor at equilibrium is calculated by the ratio between the hydrated filler volume (V) and initial filler volume (V0) (swelling factor = V/V0) [11]. Because the swelling properties are due to the insoluble HA in hydrogel, the hydration capacity may be calculated using the water-insoluble HA in each formulation (mg). In this case, the insoluble HA’s hydration capacity is given by the ratio between V and insoluble HA in the gel (mg) [61]. The higher the swelling factor, the further away the gel is from equilibrium. An HA filler close to its equilibrium has already reached its hydration capacity and then has a low swelling factor; a filler far from its equilibrium has a higher swelling factor and hydration capacity. All crosslinked HA fillers absorb added solvent and swell, but they can swell only to a certain level as is determined by the polymer network thickness and by the grade of crosslinking [13,65,66]. The crosslink grade determines the capability of the polymer chains to remain bonded together, limiting the chains from outdistance, reducing water molecule penetration and binding. Consequently, the fillers with high G′ have a lower fluid uptake capacity due to their tighter gel network and related lower gel’s ability to expand [49]. A filler with a low swelling factor after injection in soft tissue has a low capacity to bind water molecules and will swell less compared to a filler with a higher swelling factor [13]. La Gatta et al. related the swelling factor to the filler’s hydro-action, which is the expression of its capacity to expand after injection owing to the binding with water molecules. The expansion after injection is desired only to a certain extent and, if excessive, could lead to undesired effects, such as palpability and edema, which may be challenging to the injectors [10].

## 3. Important Factors When Using Hyaluronic Acid Fillers for Dermatological Surgery

We identify four different clinical stages that characterize the injection of an HA filler: the injection, tissue integration, volume restoration, and hydration stages. In each of these phases, specific rheological and biophysical characteristics in the fillers may modify the behavior of filler during the injection, the initial integration, and the long-lasting stay in soft tissue.

### 3.1. The Injection Phase

This phase is composed of different steps: the passage through the needle and the immediate integration into the soft tissues. The rheological feature that is an expression of injectability through the needle is complex viscosity. Major filler producers provide needle sizes of 30 G or 27 G, with a few volumizing fillers needing a 25 G or a 23 G needle. Adequate viscosity is necessary to allow the injection through fine needles and to maintain a good capacity of volume restoration in the filler. To achieve this balance, manufacturers have developed specific technologies and formulations for their HA fillers. They must possess sufficient elasticity but also sufficiently low viscosity to be extruded through a thin needle. The lower the viscosity is, the lower extrusion force is needed by the injector to push the filler through the syringe. Another useful parameter to describe the filler during this phase is the yield stress. The yield stress indicates the shear force intensity that determines the disruption in the filler’s physicochemical structure and loss of viscoelasticity. Low yield stress increases the risk for the filler to be subject to excessive shear deformation and undergo a structural alteration during injection through a fine needle. Moreover, a filler with a low η* also has a low shear thinning point and is easily injected because it does not need a high pressure on the syringe plunger.

### 3.2. Tissue Integration

The filler, immediately after the extrusion through the needle, enters and spreads into the soft tissue. This step decides the initial integration of the filler and is influenced by some gel characteristics, such as viscosity, tan δ, and cohesivity. The integration of the filler into tissue is a relevant aspect because it has an important role in the final correction of defects and in the homogenous distribution of the filler. A good and homogenous integration reduces the risks of nodules due to material accumulation. Sundaram et al. [14] stated that a low-η* HA filler tends to spread more homogenously and has a softer feel and reduced palpability. Viscosity is a rheological feature that can be used to identify the correct injection plane of fillers in the dermis. This skin layer is firmer than fatty tissue and, inside, it can spread only fillers with low η*. The lower the η*, the more superficial the injected dermal layer, while in the subcutaneous layer, η* is less important because of lower tissue consistency. On the other hand, it is important to remember that the fillers with low η* have a low elasticity (G′) and, therefore, a low lifting or volume restoration capacity. For this reason, fillers with low η* will be indicated for superficial injection and for fine-line correction and some fillers with very low η* need to be injected in the superficial dermis using specific techniques as a “blanching technique” [67].

In addition, tan δ indicates the spreadability of the filler in the soft tissues. A filler with a high tan δ (close to 1) has a high viscous component and a low elastic component. A more “liquid gel” can spread within the dermal collagen fibrils network more easily than an “elastic gel” (Figure 3) [68]. Monophasic fillers have been described with tan δ that is inversely proportional to G′; this may be a parameter that indicates how close a filler is to the liquid gel condition [31,69].

Although debated, the role of the cohesivity represents another key element in the description of filler/soft tissue integration. Some authors [32,68] have compared the histology of the dermis after injection of different HA fillers with different viscoelastic characteristics. They describe different distribution patterns in dermal and subdermal tissues. Fillers with higher cohesivity tend to have less clumping and more homogenous staining and are evenly placed in and between the collagen fibers throughout the reticular dermis. Fillers with lower cohesivity produce large pools of HA distributed as clumps or beads of the material located at the lower portion of the dermis, with the upper and mid reticular dermis being free of material. Fillers with intermediate cohesivity show an intermediate histological pattern. These results conflict with the definition of cohesivity as “the force between particles of the same substance that acts to unite them”. The likely explanation for the different histological distribution is related more to viscosity and elasticity than it is to cohesivity [53].

The discordance concerning the role of cohesivity in tissue integration exists probably because there are no standardized methodologies to calculate this parameter.

### 3.3. Volume Restoration

The main action in HA fillers, particularly the volumizing types, is the volume restoration or lifting of soft tissues. Consequently, the rheological features that describe this capacity are the main characteristics in the fillers. The lifting capacity is the HA filler’s ability to lift tissue and resist deformation after the injection. The elastic moduli are the measures commonly recognized to describe this action. The more useful modulus in this discussion is G′ because the surrounding tissue has more shear than vertical forces due to gravity and the action of facial muscles. In general, the higher the G′, the higher the elasticity and resistance of the filler to deformation and, ultimately, its capability to restore the soft tissue volume [11,47,49]. Hee et al. [40] stated that G′ had a positive correlation to the overall lift capacity when fillers of similar composition/crosslinking technology were compared (i.e., only HA monophasic fillers or only HA biphasic fillers). HA fillers with higher G′ must be injected into the deep-fat compartments or the pre-periosteal plane because these fillers have good volume-restoration properties. As a rule, these fillers must be injected deep into the soft tissues where they are neither palpable nor visible. The HA fillers with intermediate G′ are injected into the superficial fat compartments, into the derma-subcutaneous junction, or deep dermal layer. Finally, the HA fillers with low G′ are injectable into the dermis.

After its injection and during its dwell into the soft tissues of the face, the HA filler is subjected not only to shear stress, but also vertical compression and stretching forces, which lead to filler deformation. For this reason, G′ is not sufficient on its own to express the filler’s volume-restoration capacity. E′ has an important role, even if it is not frequently used in rheological studies. An essential aspect of volume enhancement in the deep-fat compartments is increasing projection. Achieving a good volume restoration in this fat layer is fundamental to increase the tissue projection in a vertical direction on the bony plane. This action is assured by the filler capacity to stay compacted in a unique conglomerate and not being divided, under vertical pressure, into multiple small amounts of filler particles. If the fillers split, they will quickly lose projection capacity and, in effect, have a decreased volumizing effect. Cohesivity, together with G′, indicates the volumizing capacity in the deep layers by the harder HA fillers and it is essential in the choice of fillers, especially among volumizing fillers. Fillers implanted into deep anatomic layers of the face are constantly subjected to compression forces and tension from outside forces. Fillers with high cohesivity can resist vertical compression and have a greater capacity to maintain their original shape after injection. Moreover, the consistency of fatty tissue in deep-fat compartments is less thick than the dermis and, therefore, in the fatty tissue, fillers can spread easier than in the dermis. The importance of cohesivity comes into light if the filler is injected into fatty tissue rather than in the dermis. Several studies [31,49,53] presented data suggesting that as G′ decreases, the gel exhibits more cohesive properties. Hence, when comparing the lasting effects of two HA fillers with the same G′ but different cohesivity, the filler with lower cohesivity loses tissue projection more easily than fillers with higher cohesivity. Cohesivity and G′ are both related to the capability of soft tissue volume restoration in the fillers and there appears to be a relationship between them, where one compensates for the lack of the other. Again, the lack of a unique and standardized methodology to determine cohesivity may be a challenge in defining its role in tissue lifting.

### 3.4. Hydration

After injection, HA filler molecules draw water in and this capacity is responsible for the restoration of a high level of tissue hydration, contributing to volume augmentation. The swelling factor is related to the insoluble HA in the filler and the ratio of non-crosslinked insoluble HA may influence the water affinity. The higher the insoluble HA concentration, the higher the swelling factor, the higher the filler volume enhancement due to water binding, and the higher the hydration of the surrounding tissue. Swelling factor seems to also be related to G′, since fillers with higher G′ have a lower swelling factor and vice versa. This is because G′ is defined by a high-crosslinking degree that prevents water penetration and binding. Selection of the proper filler in areas prone to edema, such as the tear trough and lips, must consider the balance between these two parameters. In the tear trough, we need a soft filler with a low G′ but with a low swelling factor. Unfortunately, these two characteristics are seen together in the same filler and this explains why it may be difficult to identify an ideal filler for this specific area.

## 4. Conclusions

The rheological characteristics influence the integration between the HA filler and the surrounding soft tissue and determine the HA filler capacity to modify the volume of the injected anatomical layer. It is mandatory for clinicians to be able to select the filler with appropriate characteristics to achieve the desired final clinical results. The filler selection is based on the anatomy of the injected area, on tissue consistency, on tissue thickness, on the tightening of the retaining areas, on the intensity and strength of mimetic muscle activity, and lastly, on external forces acting on the facial zone and on the anatomical layer selected by the injector. In each facial area, all these variables change and the selection of filler necessitates an accurate assessment. Other considerations that may come into play when selecting fillers include the presence or absence of local anesthetic. Since this has become standard practice in filler manufacturing, future studies may explore the effects of lidocaine on the physicochemical properties of fillers upon integration in soft tissue.

Degradation of fillers with the use of hyaluronidase is an important safety aspect of HA. HA products, particularly crosslinked ones, are easily degraded, regardless of their rheology or physicochemical properties and/or manufacturing technologies [70]. This information is crucial for clinicians who may prefer certain brands to ensure safe filler treatments. Lastly, practitioners who wish to impart longer-lasting lifting and rejuvenating effects may opt for fillers that could produce prominent projection and have long-term results, with minimal amounts of material (e.g., fillers with XTR™ technology).

## Figures and Tables

**Figure 1 ijms-23-10518-f001:**
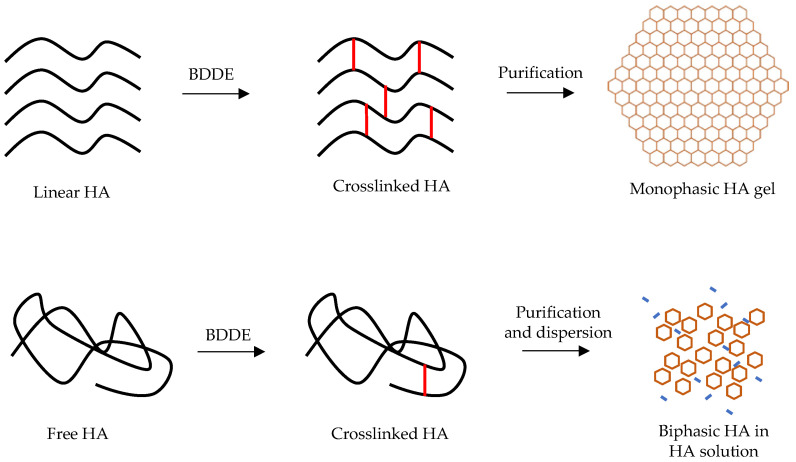
Monophasic vs. biphasic fillers.

**Figure 2 ijms-23-10518-f002:**
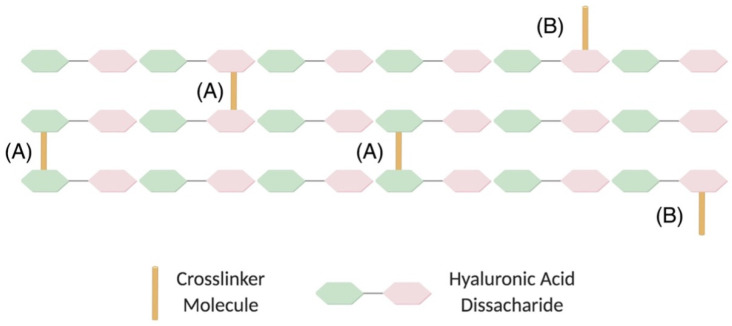
Diagram of HA crosslinking through (A) a crosslinker molecule linking two chains and (B) a crosslinker linked to only one chain. HA, hyaluronic acid (Reprinted from Ref. [25]).

**Figure 3 ijms-23-10518-f003:**
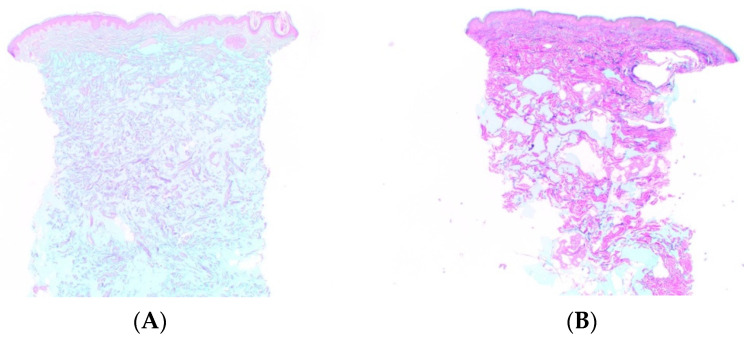
Histologic findings in superficial and mid-reticular dermis in (**A**) a liquid gel (CPM technology) vs. (**B**) a more elastic gel (Vycross) at 15 days post-injection. Magnification ×12.5. Reprinted/adapted with permission from Ref. [68]. Copyright 2017, Matrix Medical Communications.

**Table 1 ijms-23-10518-t001:** A summary of filler rheological and physicochemical characteristics and their clinical implications.

Rheological Characteristics	
Storage/Elastic modulus (G′)	It measures the energy stored by the gel during deformation and is used to recover its original shape. It represents the elastic behaviour of a gel or how much it can recover its shape after shear deformation. Unit of measurement: pascal (Pa).
Loss/Viscous modulus (G″)	It measures the energy lost on shear deformation through internal friction. It represents the inability of the gel to recover its shape completely after shear deformation. Unit of measurement: pascal (Pa)
Complex modulus (G*)	It measures the total energy needed to deform material using shear stress. It represents how difficult it is to alter the shape of an individual crosslinked unit of filler. Unit of measurement: pascal (Pa)
Tangential delta (tan δ)	It is a measure of the ratio of viscous to elastic components of G*, defined as tan δ = G″/G′ It refers to the elasticity of a filler. Measures whether a filler is more elastic (gel-like) or more viscous (liquid-like).
Cohesivity	It describes the internal adhesion forces holding together individual crosslinked HA units. It is an indicator of the filler’s resistance to vertical compression or to stretching. Three different methods and units of measurement are available: (a) linear compression test, (b) average drop-weight (c) Gavard-Sundaram Cohesivity Scale. Each method has a specific unit of measurement.
Complex viscosity (η*)	It is a measure of the resistance to deformation when shear stress is applied [9]. For fillers, it corresponds to the concept of “thickness” or of “resistance to flow” during the injection. Unit of measurement: pascal-second (Pa·s)
**Physicochemical characteristics**	
Swelling ratio (mL/g) or hydration capacity (mL/mL0)	It describes the HA filler water uptake capability expressed by the volume that a determined quantity of the biopolymer contained in the filler or a determined volume of gel could reach when incubated with phosphate-buffered saline [10,11,12,13]. It is correlated with the thickness of the polymer network, with crosslinking grade, with HA concentration. Unit of measurement: mL/g or mL/mL0 (mL is volume of the fully swollen gel and mL0 is initial gel volume)
HA concentration (mg/mL)	It indicates the mg of HA in 1 mL of gel, including both nonextractable (insoluble) and extractable (soluble) HA. The insoluble HA is the crosslinked component that provides long-lasting filler’s presence in the soft tissue. The soluble HA is the not crosslinked part that is easily metabolized and does not contribute to the extended duration.
Degree of modification and degree of crosslinking	The degree of modification is the stoichiometric ratio between the sum of mono- and double-linked BDPE residues and HA disaccharide units. It provides the total amount of linked BDDE in comparison to total amount of HA. It describes the total change in the polymer after modification. It is indicated in percentage (%). The degree of crosslinking is the stoichiometric ratio between BDDE residues that are double-linked and HA disaccharide units. It is indicated in percentage (%). The crosslinking ratio is the ratio of crosslinked BDDE to the total number of BDDE molecules bound to HA.

**Table 2 ijms-23-10518-t002:** Select filler brands, manufacturers, and crosslinking technology [28,29,30,31].

Filler’s Brands (Europe)	Company	Crosslinking Technology	Notes
Juvederm Ultra 2,3,4	Allergan	Hylacross technology	Lasts 12 months; contain a high ratio of high MW HA vs. lower MW HA
Juvederm Volux, Voluma, Volift, Volbella, Volite	Allergan	Vycross technology	Lasts up to 18 months; contain a higher proportion of low MW HA vs. higher MW HA
Saypha Filler, Volume, Volume Plus	Croma	Supreme Monophasic and Reticulated Technology (SMART)	-
Restylane Vital, Vital light, Restylane, Restylane Lyft	Galderma	Non-animal stabilized hyaluronic acid technology (NASHA)	Lasts 6 months on average, with retreatment every 6–9 months; addition of small amounts of BDDE introduces minute amounts of crosslinks between the individual chains, leading to entangled matrix
Restylane Fynesse, Refyne, Kysse, Volyme, Defyne	Galderma	Optimal balance technology (OBT)	Thicker or thinner fillers are obtained by varying gel calibration, and firmer or softer fillers by varying crosslinking
Yvoire Classic, Volume Contour	LGChem	High Concentration Equalized crosslinking technology (HICE)	Has a maximal rate of crosslinking, minimal alteration in HA structure, and optimization of dispersion of the crosslinking agent
Belotero Soft, Balance, Intense, Volume, Lips-Shape, Lips-Contour	Merz	Cohesive Polydensified Matrix (CPM)	Lasts up to 12 months; monophasic polydensified gel that combines high levels of crosslinked HA with lighter levels of crosslinked HA in a cohesive matrix
Definisse Fillers	RELIFE	eXcellent Three-dimensional Reticulation (XTR™) technology	A mixture of different lengths of HA chains are intermolecularly bound by a crosslinking agent to produce a stable three-dimensional HA matrix
Teosyal RHA 1, 2, 3, 4, Kiss	Teoxane	Resilient Hyaluronic Acid (RHA)	Lasts 6–9 months; produces gels with long HA chains stabilized by natural and chemical crosslinks
Stylage Hydromax, S, M, L, Lip, XL, XXL	Vivacy	Interpenetrating Network like (IPN-Like)	Use several individual crosslinked matrices, which undergo an interpenetrating network-like process to achieve a monophasic gel with an increased density of crosslinking

**Table 3 ijms-23-10518-t003:** Viscoelastic properties at 0.1 Hz (T = 37 °C) and indications of representative HA soft tissue fillers (Adapted with permission from Ref. [45]. Copyright 2011, Società Italiana Biomateriali).

Filler’s Brands (Europe)	G′ (Pa)	η (Pas)	Indications
Juvederm Ultra 3	173.28 ± 20.63	1629.90 ± 233.33	Wrinkles between nose and corner of mouth
Juvederm Ultra 4	102.21 ± 11.46	1479.10 ± 75.41	Severe folds and lines and for facial contouring
Juvederm Voluma	603.14 ± 58.34	1033.40 ± 50.37	For restoring volume loss (e.g., cheeks)
Belotero Soft	6.93 ± 0.73	149.09 ± 46.19	Fine superficial folds, including crow’s feet and perioral lines
Belotero Intense	76.41 ± 7.90	1008.70 ± 115.06	Deep folds and lip and volume augmentation
Restylane	301.08 ± 8.55	230.35 ± 61.25	Creases, wrinkles, scars, and lip enhancement

## Data Availability

Not applicable.

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
