# Peer review of "The Rheology and Physicochemical Characteristics of Hyaluronic Acid Fillers: Their Clinical Implications"

_ijms, 2022, doi:10.3390/ijms231810518_

Round 1
Reviewer 1 Report
The paper reviews rheological properties of hyaluronic acid. The subject of the paper is interesting – it would be very convenient to have characteristics of commercial hyaluronic acid products. On the other hand, it is very judge about the paper because the content does not reflect the title of the paper. The paper contains not any value, not any graph showing rheological properties.
Rheological behavior is very complicated phenomenon. Author mention complex viscosity obtained in oscillatory experiment, but have completely forgotten viscosity measured in flow (rotation and capillary) experiments. These two viscosities are different. Shear thinning point is not determined fro complex viscosity.
Viscoelastic behavior depends on amplitude and time scale of deformation. It has to be highlighted. Shear thinning has to be discussed. Thixotropy has to be discussed.
PHYSICOCHEMICAL CHARACTERISTICS shall include molecular weight and polydispersity. Rheological properties also depend on molecular weight, polydispersity and concentration. It shall be discussed.
Authors mention “hard” and “soft” fillers. What are their rheological properties? What are rheological properties of “monophasic” and the “biphasic” systems?
Table 2 contains some commercial products. What are their rheological/mechanical properties? What are values of viscosity and yield stress?
Table 2 contains the names of crosslinking technologies. What is their essence?
Author Response
Response to reviewer 1 comments
Point 1: The paper reviews rheological properties of hyaluronic acid. The subject of the paper is interesting – it would be very convenient to have characteristics of commercial hyaluronic acid products. On the other hand, it is very judge about the paper because the content does not reflect the title of the paper. The paper contains not any value, not any graph showing rheological properties.
Response 1: We appreciate this feedback. In the text, we have cited several studies which have reported comparative analysis of the rheological properties of various filler brands. One paper that we have developed in the past (Salti G, Fundarò SP. Evaluation of the Rheologic and Physicochemical Properties of a Novel Hyaluronic Acid Filler Range with eXcellent Three-Dimensional Reticulation (XTR™) Technology. Polymers (Basel) 2020 Jul 24;12(8):1644. doi: 10.3390/polym12081644. PMID: 32722003; PMCID: PMC7463506) provides an in-depth analysis of in vitro properties of hyaluronic acid fillers from different manufacturers. Our current review discusses the possible correlation of these filler properties to the performance of the various brands in vivo including when the filler is injected, as it integrates into soft tissue, and how it behaves when volumizing the face. We have planned future publications showing the effects of rheological properties (e.g., fillers manufactured via the XTR™ technology) on volumizing, skin density/thickness, elasticity, firmness, tonicity, among others. The hope is that through our current submission, major filler brands could disclose the same analysis to aid in the choice of fillers for particular indications.
Point 2: Rheological behavior is very complicated phenomenon. Author mention complex viscosity obtained in oscillatory experiment, but have completely forgotten viscosity measured in flow (rotation and capillary) experiments. These two viscosities are different. Shear thinning point is not determined from complex viscosity.
Response 2: Thank you kindly for the feedback. Complex viscosity has been widely discussed in the domain of filler rheology and appears to be the type of viscosity that has clinical significance, which was alluded to in our manuscript. Although we have discussed the types of rheometers, including capillary and rotational or shear rheometers, there is paucity of published data on soft tissue filler viscosity using capillary rheometer readings. We acknowledge that this may be a future topic of research.
Point 3: Viscoelastic behavior depends on amplitude and time scale of deformation. It has to be highlighted. Shear thinning has to be discussed. Thixotropy has to be discussed.
Response 3: Grateful for the inputs. We have highlighted the importance of amplitude and time-scale of deformation in the text. Shear thinning was discussed as it relates to injectability of the fillers. We added a short discussion on the implications of thixotropy to clinical outcomes.
Point 4: PHYSICOCHEMICAL CHARACTERISTICS shall include molecular weight and polydispersity. Rheological properties also depend on molecular weight, polydispersity and concentration. It shall be discussed.
Response 4: Thank you for this comment. We have added a few paragraphs to accommodate a discussion on molecular weight and polydispersity. A paragraph regarding concentration has already been previously included.
Point 5: Authors mention “hard” and “soft” fillers. What are their rheological properties? What are rheological properties of “monophasic” and the “biphasic” systems?
Response 5: Additional notes on the properties of hard and soft fillers and mono- and biphasic systems have been provided. Relative comparison of the rheologic properties of these filler types have been outlined.
Point 6: Table 2 contains some commercial products. What are their rheological/mechanical properties? What are values of viscosity and yield stress?
Response 6: We have added a table on viscoelastic data from a published source comparing commonly used fillers. Apparently, variability may be expected if taken from different studies due to the manner of instrumentation used in each trial.
Point 7: Table 2 contains the names of crosslinking technologies. What is their essence?
Response 7: The intention was to provide the reader a summary of the currently available proprietary formulae to equip them with the necessary information to help in selection of fillers appropriate for their patients.
Reviewer 2 Report
Summary
This is an interesting approach in the topic of hyaluronic acid as a material for clinical applications. This review article focuses on hyaluronic acid-based fillers for dermatological surgery. The authors have provided an extensive description on terms used in the physical characterization of such materials which help readers who may have little to no background in material science. Additionally, the authors have provided insights on factors to consider at different phases of the injection of the fillers into the dermis. This would be helpful towards material scientists who can provide solutions to clinical problems.
Review
Completeness of the review topic covered: Scope of the article ranges from the rheological and biophysical properties of hyaluronic acid. It also includes explanations of the common terms used to characterize hyaluronic acid as a hydrogel or as a soluble material. However, there are a few important aspects that the authors could have mentioned or have a heavier discussion on. For example, there could have been clarity on the sources of these clinical-grade hyaluronic acid and their implications and a clearer discussion on the effect of molecular weights of hyaluronic acid since they can range from 5 kDa to more than 1000 kDa.
Relevance of the review topic: IJMS is interested in fundamental problems of broad interest in biology, chemistry and medicine. Additionally, this is for a special issue titled “Hyaluronic Acid: A Versatile Polysaccharide for Biomedical Applications 2022”. Therefore, hyaluronic acid as materials for surgical fillers is relevant for this journal.
The gap in knowledge identified: There have been few articles reviewing hyaluronic acid as a filler for dermatological surgery in the past 10 years. They also identified that material characteristics and properties have been overlooked or have been poorly understood by users of these materials.
The appropriateness of references: The authors provided 56 references. Typically, review type articles contain at least 70 to 150 references. Would it be a problem to add more references to the article? In addition, the formatting of majority of the references needs attention. Many are either incomplete or journal titles are not in the correct format. This is important because readers of review articles would want to find and read the articles the authors have suggested.
The authors might want to add some figures to provide visual aids. For example, a figure to show how different crosslinks or degree of crosslinks might affect mechanical properties, or a figure to show the difference between monophasic and biphasic fillers. They may reproduce figures from the sources they have covered (with the appropriate permissions from the publishers).
Specific comments
(Page 1, Line 42): The authors may use a different heading for this section. “Results” is not an appropriate for review articles. The authors may use the past reviews published by IJMS as references for possible headings for the various sections.
(Page 2, Table 1): Please change “Unit of measure” to “Unit of measurement”.
(Page 3, Table 1): The authors used “MoD” as the acronym for “degree of modification”. But this is not common within the field and it is not intuitive why it is “MoD” instead of “DoM”. I would suggest not using the acronym at all. It is only used in this table and the paragraph on Page 4, Line 110.
(Page 3, Table 1): Similarly, for “CrD” and “CrR”. These are not common acronyms and not needed. I would suggest not using the acronym and just spell out the entire terms.
(Page 3, Line 66): What are the molecular weights of the hyaluronic acid used? Would the length of the molecule affect the material properties like viscosity? Longer molecules would be more appropriate for monophasic? Would molecular weight affect immunological reactions? The authors may organize this discussion in wherever they think would be more appropriate.
(Page 3, Line 73): Could the authors introduce in 1 paragraph where hyaluronic acid may be obtained from? What are the common and preferred sources (bacteria, rooster combs (avian) were listed on FDA website). While the hyaluronic acid has no difference between species, the extraction process might not be efficient in removing impurities. Therefore, would the small amount of impurities could result in immunological reactions?
(Page 5, Line 148): The authors provided a list of filler manufacturers in Table 2 but scientists may not be familiar with the brand names and the proprietary crosslinking technologies are hard to interpret. Therefore, could the authors provide more description of the technology? For example, a column on type of treatment or how long each filler can last in the body? Or any factors that differentiate all the fillers available in the market. If there is a practicing clinician among the authors, how do clinicians decide which manufacturer to pick for the specific treatments?
(Page 12, Line 503): The authors may use a different heading for this section. Similarly, “Discussion” is not an appropriate heading. The authors may consider a heading such as “Important factors when using hyaluronic acid fillers for dermatological surgery”.
(Page 13, Line 550): The authors might consider providing a figure to aid in the discussion on the histology of the dermis.
(Page 15, Line 621): Is this paragraph more appropriate as for “Conclusions” and future prospects? The authors should end the article with a “Conclusions” section.
(Page 15, Line 642): The authors may shift and incorporate this paragraph into the introduction. A section for “Materials and Methods” is not required for review articles”.
(Page 15, Line 641): Could the authors expand on the “XTR technology” somewhere near Table 2. While it was listed in Table 2, there was no explanation or description or critique on this. But the authors mentioned that this imparts longer lasting results. It would be helpful towards readers who are not familiar with this technology.
(Page 15, Line 647): Why was the literature search limited to December 2020? The article was submitted for peer-review in August 2022. There is a gap of 20 months. Could expanding to scope to this year (2022) help in adding more references to the review article?
(Page 16, Line 671): Please check the formatting for all of the references listed. For example, “discussion S143” shouldn’t be there. Wiley suggests citing this reference as Falcone, S.J., Doerfler, A.M. and Berg, R.A. (2007), Novel Synthetic Dermal Fillers Based on Sodium Carboxymethylcellulose: Comparison with Crosslinked Hyaluronic Acid–Based Dermal Fillers. Dermatologic Surgery, 33: S136-S143. https://doi.org/10.1111/j.1524-4725.2007.33353.x
(Page 16, Line 682): Please check the formatting of the references.
(Page 16, Line 694): Please check the formatting of the references. The journal name is missing.
(Page 16, Line 707): Please check the formatting of the references.
Author Response
Response to reviewer 2 comments
Point 1: This is an interesting approach in the topic of hyaluronic acid as a material for clinical applications. This review article focuses on hyaluronic acid-based fillers for dermatological surgery. The authors have provided an extensive description on terms used in the physical characterization of such materials which help readers who may have little to no background in material science. Additionally, the authors have provided insights on factors to consider at different phases of the injection of the fillers into the dermis. This would be helpful towards material scientists who can provide solutions to clinical problems.
Response 1: We highly appreciate your insights on our paper. Our hope is to equip injectors with valuable information regarding available fillers and their properties to help them select the appropriate fillers for specific indications for their patients.
Point 2: Completeness of the review topic covered: Scope of the article ranges from the rheological and biophysical properties of hyaluronic acid. It also includes explanations of the common terms used to characterize hyaluronic acid as a hydrogel or as a soluble material. However, there are a few important aspects that the authors could have mentioned or have a heavier discussion on. For example, there could have been clarity on the sources of these clinical-grade hyaluronic acid and their implications and a clearer discussion on the effect of molecular weights of hyaluronic acid since they can range from 5 kDa to more than 1000 kDa.
Response 2: Thank you kindly for the feedback. We have added a section on molecular weight and polydispersity. Although there is variability of molecular weight of HA among filler brands, gel characterization through defining cross-linking and degree of modification provides more relevant information for the clinician. We have also added in the concept of thixotropy and how it relates to filler selection.
Point 3: Relevance of the review topic: IJMS is interested in fundamental problems of broad interest in biology, chemistry and medicine. Additionally, this is for a special issue titled “Hyaluronic Acid: A Versatile Polysaccharide for Biomedical Applications 2022”. Therefore, hyaluronic acid as materials for surgical fillers is relevant for this journal.
Response 3: We agree with this comment and wish to provide more relevant educational materials for learning injectors.
Point 4: The gap in knowledge identified: There have been few articles reviewing hyaluronic acid as a filler for dermatological surgery in the past 10 years. They also identified that material characteristics and properties have been overlooked or have been poorly understood by users of these materials.
Response 4: Our review highlights the need for further studies elucidating the HA filler properties, including rheology and physicochemical characteristics, that may influence the choice of fillers for varying indications.
Point 5: The appropriateness of references: The authors provided 56 references. Typically, review type articles contain at least 70 to 150 references. Would it be a problem to add more references to the article? In addition, the formatting of majority of the references needs attention. Many are either incomplete or journal titles are not in the correct format. This is important because readers of review articles would want to find and read the articles the authors have suggested.
The authors might want to add some figures to provide visual aids. For example, a figure to show how different crosslinks or degree of crosslinks might affect mechanical properties, or a figure to show the difference between monophasic and biphasic fillers. They may reproduce figures from the sources they have covered (with the appropriate permissions from the publishers).
Response 5: We appreciate your feedback. For this, we have incorporated additional studies and have ensured the accuracy of formatting of the bibliography. We have also added visual aids to help the reader conceptualize crosslinking, and differentiate between monophasic and biphasic fillers.
Point 6: (Page 1, Line 42): The authors may use a different heading for this section. “Results” is not an appropriate for review articles. The authors may use the past reviews published by IJMS as references for possible headings for the various sections.
(Page 2, Table 1): Please change “Unit of measure” to “Unit of measurement”.
(Page 3, Table 1): The authors used “MoD” as the acronym for “degree of modification”. But this is not common within the field and it is not intuitive why it is “MoD” instead of “DoM”. I would suggest not using the acronym at all. It is only used in this table and the paragraph on Page 4, Line 110.
(Page 3, Table 1): Similarly, for “CrD” and “CrR”. These are not common acronyms and not needed. I would suggest not using the acronym and just spell out the entire terms.
Response 6: We thank the reviewers for their constructive feedback. We have revised the text by removing the acronyms and have reworded the appropriate text. The “Results” and “Discussion” headers have been changed to “A review of rheology and physicochemical characteristics of HA fillers” and “Important factors when using hyaluronic acid fillers for dermatological surgery,” respectively.
Point 7: (Page 3, Line 66): What are the molecular weights of the hyaluronic acid used? Would the length of the molecule affect the material properties like viscosity? Longer molecules would be more appropriate for monophasic? Would molecular weight affect immunological reactions? The authors may organize this discussion in wherever they think would be more appropriate.
Response 7: Appreciate the helpful comments. We have added a paragraph on the significance of molecular weight (lines 515 to 530).
Point 8: (Page 3, Line 73): Could the authors introduce in 1 paragraph where hyaluronic acid may be obtained from? What are the common and preferred sources (bacteria, rooster combs (avian) were listed on FDA website). While the hyaluronic acid has no difference between species, the extraction process might not be efficient in removing impurities. Therefore, would the small amount of impurities could result in immunological reactions?
Response 8: We have added a brief description regarding common HA sources and the advantages of using bacterial HA. Thank you.
Point 9: (Page 5, Line 148): The authors provided a list of filler manufacturers in Table 2 but scientists may not be familiar with the brand names and the proprietary crosslinking technologies are hard to interpret. Therefore, could the authors provide more description of the technology? For example, a column on type of treatment or how long each filler can last in the body? Or any factors that differentiate all the fillers available in the market. If there is a practicing clinician among the authors, how do clinicians decide which manufacturer to pick for the specific treatments?
Response 9: We have added available evidence on the characteristics of these proprietary crosslinking technologies from available peer-reviewed sources.
Point 10: (Page 12, Line 503): The authors may use a different heading for this section. Similarly, “Discussion” is not an appropriate heading. The authors may consider a heading such as “Important factors when using hyaluronic acid fillers for dermatological surgery”.
Response 10: This has been amended. Thank you.
Point 11: (Page 13, Line 550): The authors might consider providing a figure to aid in the discussion on the histology of the dermis.
Response 11: Thank you for this comment. We have suggested to include images from the histologic study conducted by Micheels evaluating Vycross vs. CPM technology. If acceptable, we will secure copyright clearance for these images.
Point 12: (Page 15, Line 621): Is this paragraph more appropriate as for “Conclusions” and future prospects? The authors should end the article with a “Conclusions” section.
Response 12: This has been corrected. We have reallocated the paragraph as our conclusion section.
Point 13: (Page 15, Line 642): The authors may shift and incorporate this paragraph into the introduction. A section for “Materials and Methods” is not required for review articles”.
Response 13: This has been corrected. We have reallocated the paragraph as our conclusion section.
Point 14: (Page 15, Line 641): Could the authors expand on the “XTR technology” somewhere near Table 2. While it was listed in Table 2, there was no explanation or description or critique on this. But the authors mentioned that this imparts longer lasting results. It would be helpful towards readers who are not familiar with this technology.
Response 14: A few sentences reviewing XTR technology have been included in the current draft (lines 369 to 371).
(Page 15, Line 647): Why was the literature search limited to December 2020? The article was submitted for peer-review in August 2022. There is a gap of 20 months. Could expanding to scope to this year (2022) help in adding more references to the review article?
Response 15: The paper was planned to be submitted in 2021 but was a little delayed. We have expanded the search to include articles published until June 2022.
Point 16: (Page 16, Line 671): Please check the formatting for all of the references listed. For example, “discussion S143” shouldn’t be there. Wiley suggests citing this reference as Falcone, S.J., Doerfler, A.M. and Berg, R.A. (2007), Novel Synthetic Dermal Fillers Based on Sodium Carboxymethylcellulose: Comparison with Crosslinked Hyaluronic Acid–Based Dermal Fillers. Dermatologic Surgery, 33: S136-S143. https://doi.org/10.1111/j.1524-4725.2007.33353.x
(Page 16, Line 682): Please check the formatting of the references.
(Page 16, Line 694): Please check the formatting of the references. The journal name is missing.
(Page 16, Line 707): Please check the formatting of the references.
Response 16: Thank you for the useful feedback. These have been corrected.
Round 2
Reviewer 2 Report
Questions and comments were addressed. No further comments.
Recommendation is to accept the manuscript for publication.